# Nickel-Imidazolium Low Transition Temperature Mixtures with Lewis-Acidic Character

**DOI:** 10.3390/molecules28176338

**Published:** 2023-08-30

**Authors:** Mario Martos, Isidro M. Pastor

**Affiliations:** Institute of Organic Synthesis, University of Alicante, ctra. San Vicente del Raspeig s/n, San Vicente del Raspeig, 03690 Alicante, Spain; mario.martos@ua.es

**Keywords:** low transition temperature mixtures, allylic alcohols, imidazolium salts, tetrachloronickelates, sustainable chemistry

## Abstract

Low transition temperature mixtures (LTTMs) are a new generation of solvents that have found extensive application in organic synthesis. The interactions between the components often generate highly activated, catalytically active species, thus opening the possibility of using LTTMs as catalysts, rather than solvents. In this work, we introduce a nickel-based imidazolium LTTM, study its thermal behavior and explore its catalytic activity in the solvent-free allylation of heterocycles with allylic alcohols. This system is effective in this reaction, affording the corresponding products in excellent yield without the need for additional purifications, thus resulting in a very environmentally friendly protocol.

## 1. Introduction

Over the last three decades, increasing concern about the impact of human activity in nature has led scientists to experience a shift towards more sustainable processes. Within organic chemistry, commonly used volatile organic solvents (VOS) have been a particularly important field of study as they constitute the main component of every reaction by weight [1]. Volatility, toxicity, flammability and low biodegradability are the main concerns being voiced; therefore, since the mid-1990s, the field of alternative solvents has been experiencing massive growth [2]. In this way, water, supercritical fluids, perfluorinated solvents, ionic liquids or biomass-derived substances have all been studied as potential replacements for VOS [2,3]. However, drawbacks related to the solubilization of hydrophobic compounds, reactivity, potential toxicity and the cost of operation have ultimately prevented any of these from displacing VOS as commodity solvents [2]. In the early 2000s, Abbott and coworkers introduced Deep Eutectic Solvents (DES) as a new breed of highly tunable and renewable solvents. According to their original definition, DES were combinations of two or more components which interact in a way that causes a depression in the melting point of the mixture [4]. This component-based definition was later revised by Coutinho and coworkers, who proposed a new definition of DES based on the non-ideality of the mixture, i.e., the experimental depression in the melting point being significantly more pronounced than the theoretical ideal value [5]. At present, the term Low Transition Temperature Mixtures (LTTMs) is used to refer to low-melting mixtures in general, including every type of DES [6].

LTTMs have been used as reaction media for a broad range of transformations, including cross-coupling reactions [7], C-H activation reactions [8], asymmetric organocatalysis [9,10], photochemistry [11], the addition of organometallic compounds [12,13] and the synthesis of heterocycles [14]. In addition, many LTTMs (such as those based on metal halides combined with organic salts or with organocatalytic scaffolds) are intrinsically active, serving as both promoters and reaction media [15,16]. In this sense, some LTTMs have been used in catalytic amount to promote reactions in solventless conditions, which contributes further to reducing the environmental impact of synthetic protocols [17].

In this work, we report a novel LTTM based on nickel chloride and 1-methoxycarbonylmethy-3-methylimidazolium chloride (mcmimCl). Nickel-based LTTMs have been extensively studied in several fields, such as in liquid crystals, as many of them present a long-range order visible through polarized optic microscopy (POM) [18,19]. In addition, many nickel salts exhibit thermochromism in the presence of humidity, due to the reversible formation of deep blue tetrachloronickelate(II) anions and halogen-bridged octahedral anions [20]. Regarding catalysis, the application of Ni-LTTMs to this field is somewhat limited, with only a few examples reporting their use for the oxygen evolution reaction [21], nitroarene reduction [22], dehydration of sugars [23], cross-coupling [24] and multicomponent reactions [25]. Herein, we present a study on the formation, structure and properties of nickel-mcmimCl LTTMs, as well as a study of their catalytic activity in allylation reactions using alcohols as substrates. Sustainability metrics (E-factor, stoichiometric factor, atom economy, Andraos’ reaction mass efficiency, materials recovery parameter and EcoScale) have been calculated to unbiasedly assess the environmental impact of the methodology.

## 2. Results and Discussion

Our choice of imidazolium salt was based on our previous experience with imidazolium LTTMs. We found that monocarboxy-imidazolium salts performed the best for the formation of low-melting mixtures [26], while being significantly more biodegradable than their dialkylimidazolium counterparts [27,28]. Thus, we combined mcmimCl with anhydrous nickel(II) chloride in several molar proportions, obtaining deep blue liquids which were then analyzed through differential scanning calorimetry (DSC), obtaining the traces compiled in Figure 1.

As evidenced by the traces, in most cases, the mixtures do not present any significant thermal events other than glass transitions (Figure 1, panels a–d), the exception being the 4:1 mcmimCl:NiCl_2_ mixture, which exhibits an exothermic event (onset 66.9 °C) immediately followed by an endothermic event (Figure 1e). This behavior is consistent with a supercooled mixture, in which the molten phase becomes metastable due to slow phase change kinetics [29]. This could be the reason why none of the other mixtures present any events save from glass transitions.

The striking deep blue color observed upon mixing and heating the samples reveals the presence of the tetrachloronickelate (NiCl_4_^2−^) anion, the formation of which is also the likely cause for the melting point depression of these LTTMs (Figure 2a). Tetrahalometallates of copper, nickel or cobalt are well-known for their tendency to form liquid crystals with a long-range order, which are of interest in ion-conductive and optic materials [19]. Thus, we decided to study the stoichiometric mixture (2:1 mcmimCl:NiCl_2_) in more detail.

First, we performed X-ray photoelectron spectroscopy (XPS) experiments, which confirmed the presence of the NiCl_4_^2−^ anion in the mixture (Figure 2b). From the obtained results, the interaction between the components seems to be complete and homogeneous, as both the nickel and chlorine surveys show a single species of each (Appendix A).

We then subjected the 2:1 mixture to further DSC analysis using a much slower temperature gradient of 2 °C/min for both cooling and heating to mitigate the effect of the slow phase change kinetics, obtaining the trace shown in Figure 3.

In this case, three thermal events are apparent in the trace: two overlapped endothermic events (onset 56 °C and 80 °C, respectively, with a total enthalpy of 76.4 J/g) and a subtle exothermic event at 108 °C, with an enthalpy of 16.9 J/g. This pattern is unconventional for typical LTTMs, which tend to show a single melting event. It is, however, similar to the thermal behavior reported for some liquid crystals, which often show multiple events in DSC corresponding to transitions to semi-ordered mesophases [19]. In this case, the first event would be then attributed to the fusion of the crystallized mixture—as no glass transitions are apparent during the heating cycle—which is overlapped with a much larger peak corresponding to the transition to a semi-ordered liquid state. The final event would correspond to the formation of an isocratic liquid.

Thus, we decided to study the mixture through POM upon cooling from the isocratic state at 150 °C. However, we were only able to observe the formation of crystals from the homogeneous isocratic mixture. This is probably due to the very narrow interval of temperatures in which the ordered phase exists (15 °C at a cooling rate of 2 °C/min), which may result in immediate solidification into crystals upon entering an ordered state.

At this point, we shifted our attention to potential applications of these systems in catalysis. We were particularly interested in Lewis acidic catalysis, based on our previous experience with metal-imidazolium LTTMs. Allyl compounds and indoles are biologically relevant motifs [30,31,32]. Thus, we selected the allylation of heterocycles with allylic alcohols as a benchmark to assess the efficacy of the nickel-based imidazolium salt LTTMs, taking the same reaction conditions for comparison. This reaction, which has been studied with different Lewis acidic catalysts, proceeds through an S_N_1 mechanism in which the catalytic system activates the alcohol and/or the nucleophile. Thus, indole was combined with (*E*)-1,3-diphenylprop-2-ene-1-ol, in a 1:1 molar ratio, for 2 h at 80 °C in the absence of a solvent, obtaining the results described in Table 1.

Using just 5 mol% of 2:1 mcmimCl:NiCl_2_ we observed a full conversion to the product after the reaction time had elapsed (Table 1, entry 1). In an attempt to decrease the amount of materials used, we tested the 1:1 mcmimCl:NiCl_2_ mixture as well, obtaining the same results (Table 1, entry 2). An inconvenient trait of the assayed mixtures is their marked hygroscopic character, which requires careful manipulation and storage under an inert atmosphere to avoid hydration. Thus, we considered the possibility of testing a 1:1 mixture of mcmimCl and nickel(II) chloride hexahydrate. This mixture, of an intense green color, presents thermochromic behavior arising from the displacement of the aquo ligands caused by chloride, forming deep-blue tetrachloronickelate anions. Upon cooling, the mixture transitions from blue-back to green, then slowly to yellow, due to the formation of halogen-bridged hydrated polymeric octahedral anions (Appendix A). This behavior is fully reversible and consistent with the reports in the literature [33,34]. For this particular mixture, the dehydration of the nickel begins at around 45 °C (Appendix A); thus, at the reaction temperature and considering the small amount of LTTM used, the mixture should be essentially equal to the anhydrous mixture. Indeed, we obtained the product in quantitative conversion, and due to the high efficacy of the catalyst, the use of stoichiometric reagents and the absence of solvents, we were able to isolate **1** in quantitative yield by adding a small amount of ethyl acetate (a sustainable solvent) and filtering off the LTTM (Table 1, entry 6). A visual inspection of the reaction vessel confirmed our theory, as a deep blue liquid was present at the end of the reaction. The LTTM interaction clearly has a very important role in the catalytic performance of the system, as the individual components were not able to promote the reaction to the same degree or at all (Table 1, entries 4 to 6).

Encouraged by our good results, we decided to continue assessing the catalytic performance of the system by assaying several differently substituted heterocycles, obtaining the results described in Figure 1.

In general terms, the LTTM showed an excellent performance, affording the allylation products in excellent to quantitative yields. The indoles reacted exclusively at the C-3 position and did not require any sort of protecting group. The electronic effects did not seem to significantly influence the outcome of the reaction, as differently substituted indoles were obtained in comparable yield, although some less electron-rich substrates took longer to fully react. Interestingly, even severely sterically hindered indoles reacted smoothly (Figure 1, compounds **3** and **4**). The protocol was extended to other heterocycles, which selectively reacted at the nitrogen. Triazole gave good results, as did 5-phenyltetrazole (Figure 1, compounds **10** and **11**). Carbazole, a challenging substrate, reacted smoothly to afford compound **12** in quantitative yield after filtration, as did 3,6-dichlorocarbazole (Figure 1, compound **13**). Remarkably, this nickel-LTTM was able promote the allylation of electron-rich benzenes, such as 1,3,5-trimethoxybenzene, obtaining the product in excellent yield (Figure 1, compound **14**). Compounds **1** to **12** were obtained as sufficiently pure after just filtering off the catalyst, which results in a very environmentally friendly protocol.

Recyclability is one of the most important aspects of heterogeneous catalysts and further contributes to reducing the impact of synthetic methodologies. Thus, we set up the synthesis of **1** once more. After the reaction time had elapsed, a small amount of ethyl acetate was used to remove the product, with the insoluble LTTM remaining in the vessel, which was then reloaded with fresh starting materials. In this way, we were able to reuse the catalyst three times without any loss of activity. To determine the stability of the catalyst, an additional cycle was carried out after leaving the catalyst exposed to air (at 30 °C with relative humidity of >50%) overnight. The LTTM remained in its yellow polymeric form and was still able to afford product **1** in quantitative conversion (Figure 4). A visual inspection of the vessel after the reaction confirmed a transition back to the blue tetrachloronickelate form, thus confirming the robustness of the nickel-imidazolium LTTM.

We then decided to calculate the sustainability metrics to assess the impact of the methodology and to compare it with the established protocols reported in the literature. Taking the synthesis of compound **1** as a model reaction, the E-factor was calculated and the yields (Y), atom economy (AE), reaction mass efficiency (RME_Andraos_), materials recovery parameter (MRP) and stoichiometric factors (SF) were combined into a vector magnitude ratio (VMR, range 0–1) for quick comparison (see Appendix A for the equations used and any additional considerations) [35]. In addition, the qualitative aspects of the processes were assessed by means of the EcoScale [36], obtaining the results described in Table 2 and Figure 5.

In general terms, the protocol using the 1:1 mcmimCl:NiCl_2_·6H_2_O mixture has better sustainability metrics than the comparable methodologies reported in the literature. The E-factor sits second lowest of the purification-free methods due to the use of neat conditions (Table 1, entry 1), and is firmly within the desirable range for the production of bulk chemicals [42]. The material efficiency metrics, measured by the VMR, are comparable to the rest of the methodologies, although this is mainly due to them having similar values of yields, a similar stoichiometric factor and the exact same atom economy. As these three are the largest values of the VMR, they exert a buffer effect over the calculation. Looking at the RME, our methodology sits third best overall, with a value of 20.1% (Appendix A). Regarding the EcoScale, this protocol obtains a score of 79, with the main penalty coming from the use of nickel. Nevertheless, the value is classified as excellent according to the scoring system proposed by the authors [36].

It should be mentioned that, although the methodology using bcmimCl has slightly better metrics for this particular reaction, the nickel-imidazolium LTTM catalyst results in higher yields in general (particularly noticeable with difficult substrates, such as 2-phenylindole or carbazole) [37], and its high activity enables the use of other substrates, such as 1,3,5-trimethoxybenzene, which are incompatible with the bcmimCl protocol. Consequently, the nickel-based Lewis acid catalyst described herein exhibits not only comparable metrics to the best previously reported methodologies, but also greater robustness in terms of its recyclability and activity.

## 3. Materials and Methods

All reagents and solvents used are commercially available (Acros (Waltham, MA, USA), Alfa Aesar (Haverhill, MA, USA), Fluorochem (Glossop, UK), Fluka (Freehold, NJ, USA), Merck (Darmstadt, Germany)) and were used without further purification. DSC experiments were carried out at the Thermal Analysis and Porous Solids unit of the Research Technical Services of the University of Alicante (SSTTI-UA) with TA Instruments (New Castle, DE, USA) Q250 differential scanning calorimeter. Samples were analyzed in sealed aluminum crucibles under nitrogen atmosphere. NMR spectra were recorded at the NMR Unit of the SSTTI-UA using Bruker (Billerica, MA, USA) AV300 Oxford, AV400 and Avance Neo 400 solution-state NMR spectrometers. ^1^H spectra were recorded at 300 or 400 MHz, whereas ^13^C spectra were recorded at 75 or 100 MHz. The solvent used was deuterated chloroform (CDCl_3_), with tetramethylsilane (TMS) as an internal standard. Chemical shifts (δ) are provided in ppm and coupling constants (*J*) are reported in hertz (Hz). Low-resolution mass spectra of the compounds were recorded on an Agilent (Santa Clara, CA, USA) 5973 Network mass spectrometer equipped with a 70 eV electronic impact (EI) ionization source and a quadrupolar mass detector operating in single ion monitoring (SIM) mode. Samples were introduced through an Agilent 6890N gas chromatography instrument equipped with a Technokroma TRB-5MS column (30 m × 0.25 mm × 0.25 μm), using helium as the mobile phase. Fragmentations are reported according to their mass to charge ratio (*m*/*z*), along with their relative intensity in parenthesis. XPS analyses were carried out at the X-ray unit of the SSTTI-UA with a Thermo Scientific (Waltham, MA, USA) K-Alpha instrument using a monochrome Al K-α X-ray source.

### 3.1. Procedure for the Preparation of 1-(Methoxycarbonylmethyl)-3-methylimidazolium Chloride (mcmimCl)

In a glass round bottom flask, methylimidazole (40 mmol, 3.2 mL) and methyl chloroacetate (40 mmol, 3.5 mL) were combined. The mixture was sonicated for 1 h, causing the formation of a dense white solid. The product was crushed with 10 mL of diethyl ether, which was then decanted off. After drying, 7.5 g of pure mcmimCl was obtained (98% yield).

### 3.2. Procedure for the Preparation of the McmimCl:xNiCl_2_·(6H_2_O) Mixtures

In a glass vial, mcmimCl (1 mmol, 190 mg) was combined with a suitable amount of nickel precursor. A magnet was then added and the mixture was stirred at 80 °C until homogeneous (around 1 h). For mixtures containing anhydrous NiCl_2_, argon was added to the flask to protect the LTTM from moisture.

### 3.3. General Procedure for the Allylation of Heterocycles Promoted by Ni-imidazolium LTTMs

In a glass microwave tube, Ni-LTTM (5 mol%, 5 mg (hydrated) or 4 mg (anhydrous)), (*E*)-1,3-diphenylprop-2-ene-1-ol (0.25 mmol, 53 mg) and indole (1 equivalent) were added. The mixture was stirred at 80 °C until completion (monitored by GC-MS). After the reaction was completed, 0.5 mL of ethyl acetate was used to remove the product, which was then filtered through a pad of cotton and magnesium sulfate. After washing the pad with an additional 0.5 mL, removing the solvent under vacuum afforded the pure allylation products (**1** to **12**). In the case of products **13** and **14**, some excess starting material remained, so the yields were calculated through NMR.

### 3.4. Spectral Data of Compounds ***1*** to ***14***


*(E)*-3-(1,3-Diphenylallyl)-1*H*-indole (**1**) [37,41]: Yellow oil, obtained pure, 99% yield; ^1^H NMR (400 MHz, CDCl_3_): δ_H_ = 7.97 (br s, 1H, NH), 7.52 (d, *J* = 8.1 Hz, 1H, CH_Ar_), 7.45–7.28 (m, 12H, CH_Ar_), 7.11 (ddd, *J* = 8.1, 7.1, 0.8 Hz, 1H, CH_Ar_), 6.94 (d, *J* = 1.3 Hz, 1H, CH_Ar_), 6.80 (dd, *J* = 15.9, 7.4 Hz, 1H, PhC=CH), 6.52 (br d, *J* = 15.9 Hz, 1H, C=CHPh), 5.20 (br d, *J* = 7.4 Hz, 1H, C=CCH); ^13^C NMR (100 MHz, CDCl_3_): δ_C_ = 143.5, 137.7, 136.8, 132.7, 130.7, 128.6, 128.6, 128.5, 127.3, 126.9, 126.5, 126.5, 122.8, 122.2, 119.9, 119.5, 118.8, 111.3, 46.4; MS (EI, 70 eV) *m*/*z* (%): 310 (M^+^ + 1, 24), 309 (M^+^, 100), 308 (M^+^ − 1, 39), 294 (10), 232 (36), 230 (16), 218 (16), 217 (17), 206 (28), 204 (17), 202 (8), 192 (15), 191 (16), 130 (18), 115 (17).



*(E)*-3-(1,3-Diphenylallyl)-1-methyl-1*H*-indole (**2**) [37]: Yellow oil, obtained pure, 98% yield; ^1^H NMR (400 MHz, CDCl_3_): δ_H_ = 7.52–7.49 (m, 1H, CH_Ar_), 7.45–7.24 (m, 12H, CH_Ar_), 7.09 (ddd, *J* = 8.0, 6.9, 1.1 Hz, 1H, CH_Ar_), 6.84 (s, 1H, CHN), 6.80 (dd, *J* = 16.0, 7.4 Hz, 1H, PhC=CH), 6.52 (d, *J* = 16.0 Hz, 1H, C=CHPh), 5.19 (br d, *J* = 7.4 Hz, 1H, C=CCH), 3.78 (s, 3H, CH_3_); ^13^C NMR (100 MHz, CDCl_3_): δ_C_ = 143.7, 137.6, 137.5, 132.8, 130.5, 128.6, 128.5, 127.5, 127.3, 127.3, 126.5, 126.4, 121.7, 120.1, 119.0, 117.2, 109.3, 46.3, 32.8; MS (EI, 70 eV) *m/z* (%): 324 (M^+^ + 1, 25), 323 (M^+^, 100), 322 (M^+^ − 1, 36), 247 (9), 246 (45), 244 (9), 232 (14), 231 (9), 220 (28), 219 (8), 218 (12), 217 (8), 204 (8), 192 (11), 191 (16), 144 (27), 131 (8), 122 (10), 115 (11).



*(E)*-3-(1,3-Diphenylallyl)-2-methyl-1*H*-indole (**3**) [37,41]: Yellow oil, obtained pure, 92% yield; ^1^H NMR (400 MHz, CDCl_3_): δ_H_ = 7.63 (br s, 1H, NH), 7.28–7.24 (m, 5H, CH_Ar_), 7.19–7.14 (m, 6H, CH_Ar_), 7.12–7.09 (m, 2H, CH_Ar_) 6.99 (ddd, *J* = 8.1, 7.1, 1.2 Hz, 1H, CH_Ar_), 6.88 (ddd, *J* = 8.1, 7.1, 1.2 Hz, 1H, CH_Ar_), 6.74 (dd, *J* = 15.8, 7.2 Hz, 1H, PhC=CH), 6.32 (dd, *J* = 15.8, 1.2 Hz, 1H, C=CHPh), 5.05 (d, *J* = 7.2 Hz, 1H, C=CCH), 2.24 (s, 3H, CH_3_); ^13^C NMR (100 MHz, CDCl_3_): δ_C_ = 143.6, 137.7, 135.5, 132.3, 131.7, 130.7, 128.6, 128.4, 128.4, 128.1, 127.2, 126.4, 126.2, 121.0, 119.5, 119.4, 112.9, 110.4, 45.2, 12.5; MS (EI, 70 eV) *m/z* (%): 324 (M^+^ + 1, 26), 323 (M^+^, 100), 322 (M^+^ − 1, 20), 309 (17), 308 (69), 246 (28), 244 (10), 232 (15), 231 (12), 230 (21), 220 (15), 218 (29), 217 (20), 202 (9), 192 (11), 191 (27), 144 (29), 131 (10), 130 (10), 115 (17).



*(E)*-3-(1,3-Diphenylallyl)-2-phenyl-1*H*-indole (**4**) [37]: Yellow oil, obtained pure, 98% yield; ^1^H NMR (400 MHz, CDCl_3_): δ_H_ = 8.11 (br s, 1H, NH), 7.62–7.59 (m, 2H, CH_Ar_), 7.55–7.50 (m, 3H, CH_Ar_), 7.48–7.41 (m, 6H, CH_Ar_), 7.37–7.33 (m, 4H, CH_Ar_), 7.28–7.24 (m, 3H, CH_Ar_) 7.09 (ddd, *J* = 8.1, 7.1, 1.0 Hz, 1H, CH_Ar_), 6.99 (dd, *J* = 15.8, 7.3 Hz, 1H, PhC=CH), 6.50 (dd, *J* = 15.8, 1.0 Hz, 1H, C=CHPh), 5.38 (d, *J* = 7.3 Hz, 1H, C=CCH); ^13^C NMR (100 MHz, CDCl_3_): δ_C_ = 143.6, 137.6, 136.3, 135.7, 133.0, 132.4, 131.2, 128.9, 128.7, 128.6, 128.4, 128.4, 128.1, 128.0, 127.2, 126.4, 126.2, 122.2, 121.3, 119.8, 113.9, 111.1, 45.3.; MS (EI, 70 eV) *m*/*z* (%): 386 (M^+^ + 1, 31), 385 (M^+^, 100), 384 (M^+^ − 1, 19), 341 (22), 331 (16), 309 (12), 308 (46), 306 (17), 304 (11), 295 (26), 294 (97), 292 (9), 291 (11), 280 (18), 278 (8), 230 (14), 218 (9), 217 (13), 205 (8), 204 (23), 203 (8), 202 (9), 194 (9), 193 (45), 192 (22), 191 (22), 189 (8), 178 (9), 176 (7), 165 (14), 153 (9), 152 (10), 146 (10), 115 (9).



*(E)*-3-(1,3-Diphenylallyl)-9-ethyl-1*H*-indole (**5**) [37]: Reddish oil, obtained pure, 99% yield; ^1^H NMR (300 MHz, CDCl_3_) δ_H_ = 7.95 (br s, 1H, NH), 7.50–7.30 (m, 11H, CH_Ar_), 7.14–7.11 (m, 2H, CH_Ar_), 6.94 (dd, *J* = 2.4, 0.7 Hz, 1H, CH_Ar_), 6.86 (dd, *J* = 15.8, 7.4 Hz, 1H, PhC=CH), 6.58 (d, *J* = 15.8 Hz, 1H, C=CHPh), 5.24 (d, *J* = 7.4 Hz, 1H, C=CCH), 2.93 (q, *J* = 7.6 Hz, 2H, CH_2_), 1.47 (t, *J* = 7.6 Hz, 3H, CH_3_); ^13^C NMR (100 MHz, CDCl_3_): δ_C_ = 143.6, 137.6, 135.6, 132.8, 130.6, 128.6, 128.6, 128.5, 127.2, 126.7, 126.6, 126.4, 122.4, 120.7, 119.8, 119.2, 117.7, 46.4, 24.0, 13.9.; MS (EI, 70 eV) *m/z* (%): 338 (M^+^ + 1, 26), 337 (M^+^, 100), 336 (M^+^ − 1, 32), 308 (23), 281 (10) 261 (7), 260 (34), 235 (7), 234 (27), 231 (9), 230 (19), 218 (9), 217 (13), 204 (9), 192 (13), 191 (21), 158 (17), 115 (11).



*(E)*-4-Bromo-3-(1,3-diphenylallyl)-1*H*-indole (**6**) [37]: Faint yellow oil, obtained pure, 98% yield; ^1^H NMR (400 MHz, CDCl_3_): δ_H_ = 8.08 (br s, 1H, NH), 7.43–7.25 (m, 12H, CH_Ar_), 7.07–7.03 (m, 1H, CH_Ar_), 6.94 (d, *J* = 2.0 Hz, 1H, CH_Ar_), 6.81 (dd, *J* = 15.9, 6.6 Hz, 1H, PhC=CH), 6.32 (dd, *J* = 15.9, 1.4 Hz, 1H, C=CHPh), 5.99 (d, *J* = 6.6 Hz, 1H, C=CCH); ^13^C NMR (100 MHz, CDCl_3_): δ_C_ = 144.0, 137.9, 137.7, 133.9, 130.8, 129.1, 128.6, 128.3, 127.2, 126.4, 126.3, 125.0, 124.9, 124.5, 123.1, 119.2, 114.6, 110.7, 44.9.; MS (EI, 70 eV) *m/z* (%): 390 (M^+^ + 3, 25), 389 (M^+^ + 2, 99), 388 (M^+^ + 1, 43), 387 (M^+^, 100), 386 (M^+^ − 1, 19), 312 (25), 311 (8), 310 (30), 309 (13), 308 (40), 307 (11), 306 (14), 298 (9), 296 (9), 286 (23), 284 (27), 231 (17), 230 (38), 229 (7), 228 (9), 217 (36), 210 (21), 208 (26), 205 (10), 204 (29), 203 (12), 202 (14), 192 (36), 191 (37), 189 (8), 177 (8), 176 (10).



*(E)*-3-(1,3-Diphenylallyl)-5-methoxy-1*H*-indole (**7**) [37]: Brownish oil, obtained pure, 99% yield; ^1^H NMR (300 MHz, CDCl_3_): δ_H_ = 7.98 (br s, 1H, NH), 7.43–7.25 (m, 11H, CH_Ar_), 6.90–6.87 (m, 3H, CH_Ar_), 6.77 (dd, *J* = 15.8, 7.3 Hz, 1H, PhC=CH), 6.50 (d, *J* = 15.8 Hz, 1H, C=CHPh), 5.13 (d, *J* = 7.3 Hz, 1H, C=CCH), 3.76 (s, 3H, OCH_3_); ^13^C NMR (75 MHz, CDCl_3_): δ_C_ = 153.8, 143.4, 137.5, 132.5, 131.9, 130.6, 128.5, 128.5, 128.5, 127.3, 127.2, 126.4, 126.3, 123.5, 118.3, 112.2, 111.9, 101.8, 55.8, 46.3; MS (EI, 70 eV) *m/z* (%): 340 (M^+^ + 1, 25), 339 (M^+^, 100), 338 (M^+^ − 1, 32), 324 (11), 308 (11), 263 (9), 262 (34), 253 (11), 236 (25), 209 (12), 204 (10), 192 (16), 191 (26), 160 (17), 147 (9), 115 (8).



*(E)*-3-(1,3-Diphenylallyl)-5-formyl-1*H*-indole (**8**) [37]: Faint yellow oil, 99% yield; ^1^H NMR (400 MHz, CDCl_3_): δ_H_ = 9.92 (s, 1H, CHO), 8.92 (br s, 1H, NH), 7.99 (m, 1H, CH_Ar_), 7.78 (dd, *J* = 8.5, 1.5 Hz, 1H, CH_Ar_), 7.46–7.25 (m, 11H, CH_Ar_), 7.06 (dd, *J* = 2.2, 0.7 Hz, 1H, CH_Ar_), 6.77 (dd, *J* = 15.8, 7.3 Hz, 1H, PhC=CH), 6.49 (d, *J* = 15.8 Hz, 1H, C=CHPh), 5.21 (d, *J* = 7.3 Hz, 1H, C=CCH); ^13^C NMR (100 MHz, CDCl_3_): δ_C_ = 192.9, 142.9, 140.4, 137.3, 132.0, 131.1, 129.2, 128.7, 128.6, 128.5, 127.5, 126.8, 126.4, 125.7, 124.7, 122.3, 120.6, 112.1, 46.0.; MS (EI, 70 eV) *m/z* (%): 338 (M^+^ + 1, 32), 337 (M^+^, 100), 336 (M^+^ − 1, 26), 308 (22), 260 (25), 234 (9), 218 (14), 217 (8), 209 (15), 204 (25), 193 (21), 192 (35), 191 (31), 115 (21).



*(E)*-3-(1,3-Diphenylallyl)-5-fluoro-1*H*-indole (**9**) [37,41]: Faint yellow oil, obtained pure, 99% yield; ^1^H NMR (400 MHz, CDCl_3_): δ_H_ = 7.96 (br s, 1H, NH), 7.46–7.26 (m, 12H, CH_Ar_), 7.15 (dd, *J* = 9.8, 2.5 Hz, 1H, CH_Ar_), 7.01–6.97 (m, 2H, CH_Ar_), 6.78 (dd, *J* = 15.8, 7.4 Hz, 1H, PhC=CH), 6.53 (d, *J* = 15.8 Hz, 1H, C=CHPh), 5.13 (d, *J* =7.3 Hz, 1H, C=CCH); ^13^C NMR (100 MHz, CDCl_3_): δ_C_ = 157.7 (d, *J* = 234.5 Hz), 143.1, 137.5, 133.3, 132.2, 130.9, 128.6, 128.5, 127.4, 127.3 (d, *J* = 9.8 Hz), 126.7, 126.5, 124.5, 118.9 (d, *J* = 4.7 Hz), 111.9 (d, *J* = 9.7 Hz), 110.6 (d, *J* = 26.5 Hz), 104.9 (d, *J* = 23.6 Hz), 46.3; MS (EI, 70 eV) *m/z* (%): 328 (M^+^ + 1, 24), 327 (M^+^, 100), 326 (M^+^ − 1, 35), 312 (11), 250 (31), 249 (9), 248 (16), 236 (18), 235 (16), 224 (28), 223 (7), 222 (16), 192 (20), 191 (22), 148 (21), 115 (17).



*(E)*-1-(1,3-Diphenylallyl)-1*H*-1,2,4-triazole (**10**) [37]: Colorless oil, obtained pure, 96% yield; ^1^H NMR (400 MHz, CDCl_3_): δ_H_ = 8.17 (s, 1H, CH_Ar_), 8.05 (s, 1H, CH_Ar_), 7.45–7.28 (m, 10H, CH_Ar_), 6.70 (dd, *J* = 15.8, 7.0 Hz, 1H, PhC=CH), 6.54 (d, *J* = 15.8 Hz, 1H, C=CHPh), 6.21 (br d, *J* = 7.0 Hz, 1H, C=CCH); ^13^C NMR (100 MHz, CDCl_3_): δ_C_ = 152.1, 142.7, 137.7, 135.6, 134.8, 129.2, 128.8, 128.8, 128.6, 127.5, 126.9, 125.7, 66.2; MS (EI, 70 eV) *m/z* (%): 261 (M^+^, 26), 233 (17), 206 (11), 193 (25), 192 (70), 191 (59), 190 (9), 189 (14), 178 (20), 165 (14), 157 (14), 146 (11), 145 (100), 144 (17), 130 (7), 117 (17), 116 (8), 115 (59), 91 (22), 89 (10), 77 (11).



*(E)*-2-(1,3-Diphenylallyl)-5-phenyl-2*H*-tetrazole (**11**) [37]: Colorless oil, obtained pure, 96% yield; ^1^H NMR (400 MHz, CDCl_3_): δ_H_ = 8.25–8.23 (m, 2H, CH_Ar_), 7.54–7.32 (m, 13H, CH_Ar_), 6.97 (dd, *J* = 15.8, 7.7 Hz, 1H, PhC=CH), 6.80–6.71 (m, 2H, HC=CH); ^13^C NMR (100 MHz, CDCl_3_): δ_C_ = 165.3, 137.2, 135.6, 135.1, 130.4, 129.1, 128.9, 128.8, 128.7, 127.53, 127.5, 127.0, 124.9, 69.9; MS (EI, 70 eV) *m/z* (%): 283 (19), 282 (90), 267 (20), 265 (14), 252 (9), 205 (31), 204 (37), 203 (44), 202 (35), 192 (13), 191 (100), 190 (11), 189 (19), 178 (16), 165 (18), 152 (7), 126 (10), 91 (8).



*(E)*-9-(1,3-Diphenylallyl)-9*H*-carbazole (**12**) [37]: Yellow oil, obtained pure, 99% yield; ^1^H NMR (400 MHz, CDCl_3_): δ_H_ = 8.23 (dt, *J* = 7.7, 0.9 Hz, 2H, CH_Ar_), 7.48–7.28 (m, 16H, CH_Ar_), 7.03 (dd, *J* = 15.8, 7.0 Hz, 1H, PhC=CH), 6.72–6.68 (m, 2H, C=CHPh, C=CCH); ^13^C NMR (100 MHz, CDCl_3_): δ_C_ = 140.2, 139.1, 136.3, 134.2, 128.8, 128.7, 128.2, 127.9, 127.3, 126.8, 125.9, 125.7, 123.7, 120.4, 119.3, 110.5, 59.9; MS (EI, 70 eV) *m/z* (%): 359 (M^+^, 8), 194 (18), 193 (100), 192 (40), 191 (29), 189 (13), 178 (19), 167 (61), 166 (18), 165 (13), 140 (8), 139 (10), 115 (44), 91 (7).



(*E*)-3,6-dichloro-9-(1,3-diphenylallyl)-9*H*-carbazole (**13**): White oil, obtained with traces of 3,6-dichlorocarbazole, 99% NMR yield; ^1^H NMR (400 MHz, CDCl_3_): δ_H_ = 8.08–8.07 (m, 2H, CH_Ar_), 7.44–7.28 (m, 14H, CH_Ar_), 6.93 (dd, *J* = 15.7, 6.9 Hz, 1H, PhC=CH), 6.64–6.59 (m, 2H, C=CHPh, C=CCH); MS (EI, 70 eV) *m*/*z* (%): 431 (M^+^ + 4, 14), 430 (M^+^ + 3, 22), 429 (M^+^ + 2, 72), 428 (M^+^ + 1, 41), 427(M^+^, 100), 426 (18), 394 (9), 393 (9), 392 (26), 356 (8), 341 (7), 327 (7), 322 (8), 314 (12), 253 (20), 237 (11), 235 (15), 193 (41), 192 (61), 191 (50), 189 (7), 178 (19), 165 (10).



(*E*)-2-(1,3-diphenylallyl)-1,3,5-trimethoxybenzene (**14**) [43]: Colorless oil, obtained with leftover 1,3,5-trimethoxybenzene, 96% NMR yield; ^1^H NMR (400 MHz, CDCl_3_): δ_H_ = 7.49 (m, 2H, CH_Ar_), 7.38–7.22 (m, 8H, CH_Ar_), 7.05 (dd, *J* = 15.8, 8.6 Hz, 1H, PhC=CH), 6.60 (d, *J* = 15.8 Hz, 1H, C=CHPh), 6.26 (s, 2H, CH_Ar_), 5.54 (d, *J* = 8.6 Hz, 1H, C=CCH), 3.88 (s, 3H, OCH_3_), 3.80 (s, 6H, 2xOCH_3_); MS (EI, 70 eV) *m*/*z* (%): 361 (M^+^ + 1, 28), 360 (M^+^, 100), 330 (20), 329 (75), 283 (14), 269 (23), 255 (7), 254 (19), 252 (8), 251 (8), 239 (13), 238 (18), 193 (14), 192 (66), 191 (42), 182 (11), 181 (83), 179 (7), 168 (12), 167 (7), 165 (11), 141 (9), 121 (8), 115 (13), 91 (25).


## 4. Conclusions

To summarize, we have presented a series of novel nickel chloride:carboxy-imidazolium LTTMs. The formation of the tetrachloronickelate anion was determined visually and confirmed by XPS analysis, with it being the reason for the melting point depression of these mixtures. The thermal behavior of the mixtures was analyzed through DSC, observing the formation of metastable glass phases at the standard 5 °C/min temperature gradients. Careful analysis revealed evidence of a more complex phase system than that of a regular LTTM, which could be compatible with a liquid crystal. Then, the system was applied as a catalyst for the allylation of heterocycles with (*E*)-1,3-diphenylprop-2-ene-1-ol in solvent-free conditions, finding that the catalytic LTTM could be generated in situ from its much more convenient hydrated form without a loss of activity. A series of differently substituted indoles as well as other heterocycles and electron-rich arenes were successfully allylated, obtaining the products in excellent yields without the need for purification in most cases, and the catalyst could be successfully recycled four times without a loss of its activity. To finish, we calculated the sustainability metrics to unbiasedly assess the impact of this methodology and compared it with the protocols described in the literature. We obtained an E-factor of 3.9, a VMR of 0.771 and an EcoScale score of 79, which was consistently within the best scoring methodologies, thus proving the sustainability of the protocol. Given the characteristics (in terms of efficiency, activity, sustainability) of using this type of LTMM as catalysts, it will be of interest to continue exploring synthetic applications in other reactions, as well as with other nucleophiles and electrophiles.

## Data Availability

The data presented in this study are available in Appendix A.

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
