# Peer review of "Nickel-Imidazolium Low Transition Temperature Mixtures with Lewis-Acidic Character"

_molecules, 2023, doi:10.3390/molecules28176338_

Round 1
Reviewer 1 Report
I did not find the reason for choosing the temperature of 80 oC for my reaction conditions. They investigated and presented it as better performing the catalyst used in different temperature conditions and with different molar ratios in the article.
In your manuscript, I could not find the optimal limit or the appropriate range for VMR.
This manuscript requires minor editing for grammar, Spelling, and grammar and style.
Reviewer 2 Report
The manuscript entitled: "Low transition temperature nickel-imidazolium mixtures with Lewis-acid character" could be recommended for publication in Molecules after some improvements: could be useful some experiments, to better describe the importance of LTTMs in preparative chemistry and in the challenging developments of new reaction media/catalysts for this kind of reactions. Moreover, it could be possible to know some limitations and/or a broad scope of substrates, increasing the tools for synthetic organic chemists.
Some questions/considerations regarding this work:
1) Could the molar ratio of indole (or heterocycles) to (E)-1,3-diphenylprop-2-ene-1-ol be described in Table 1, Table 2 and within the text? A comment might be needed when the reaction outcome is different in the presence of different amounts of reactant;
2) LTTM require strictly storage under inert atmosphere to avoid hydration, or it could be possible to store the mixture just under vacuum ?
3) Are the 3:1 and 4:1 mcmimCl:NiCl2 mixtures also useful catalyst for the reaction of Indole and (E)-1,3-diphenylprop-2-ene-1-ol?
4) Is the temperature of 80 °C necessary to melt the starting material? Could be used a lower temperature with liquid reagents instead?
5) Could be possible to describe an hypothetic catalytic cycle? Could the reaction proceed via a SN1 pathway? What happens in a reaction with enantiopure (E)-1,3-diphenylprop-2-ene-1-ol allylic alcohol? Is a racemic product observed after the reaction? Is it possible to check if an allyl isomerization occurs, by using a (E)-1-aryl-3-phenylprop-2-ene-1-ol?
6) In view of the reaction selectivity of indole and/or other heterocycles, could it be possible, in principle, to use an enamine as a nucleophile? Is the beta-carbon of N-C=C involved in the reaction in this case?
7) Regarding electron-rich aromatic compounds, is it possible to react aniline and (E)-1,3-diphenylprop-2-ene-1-ol under the optimized reaction conditions?
8) Regarding the synthetic utility of allyl derivatives, could the reaction also work well with purine or its analogues?
9) What is the highest number of consecutive cycles that can be done with 5% catalyst? Is it possible to detect (and quantify) if there is the catalyst leaching after each cycle?
10) Is it possible to observe good conversions with a lower amount of catalyst? Is it possible to recover the active catalyst at the end of the reaction also in this case?
11) In the case of products 13 and 14, which internal standard was used for the NMR calculation of the yields?
12) In the description of some experimental data, the use of "m" should be preferred for second order spectra, where the magnetic non-equivalence of the nuclei cannot be described through identical coupling constants, as usual for first order spectra.
Some references should be updated, also in the experimental section for previously reported compounds:
10.1039/D3GC01017A; 10.1021/acs.chemrev.8b00306; 10.1021/acs.jmedchem.2c01406; 10.1039/C6QO00227G; 10.1002/asia.201000651; 10.1055/sos-SD-047-00234.
Author Response
Please see the attachement.
